# Simultaneous Extraction of Planetary Boundary-Layer Height and Aerosol Optical Properties from Coherent Doppler Wind Lidar

**DOI:** 10.3390/s22093412

**Published:** 2022-04-29

**Authors:** Yehui Chen, Xiaomei Jin, Ningquan Weng, Wenyue Zhu, Qing Liu, Jie Chen

**Affiliations:** 1Key Laboratory of Atmospheric Optics, Anhui Institute of Optics and Fine Mechanics, HFIPS, Chinese Academy of Sciences, Hefei 230031, China; yehuich@mail.ustc.edu.cn (Y.C.); xmjin@aiofm.ac.cn (X.J.); zhuwenyue@aiofm.ac.cn (W.Z.); liuqing@aiofm.ac.cn (Q.L.); jiechen@mail.ustc.edu.cn (J.C.); 2Science Island Branch of Graduate School, University of Science and Technology of China, Hefei 230026, China; 3Advanced Laser Technology Laboratory of Anhui Province, Hefei 230037, China

**Keywords:** planetary boundary layer (PBL), aerosol extinction coefficient (AEC), aerosol optical depth (AOD), wavelet covariance transform (WCT), dilation operation

## Abstract

Planetary boundary-layer height is an important physical quantity for weather forecasting models and atmosphere environment assessment. A method of simultaneously extracting the surface-layer height (SLH), mixed-layer height (MLH), and aerosol optical properties, which include aerosol extinction coefficient (AEC) and aerosol optical depth (AOD), based on the signal-to-noise ratio (SNR) of the same coherent Doppler wind lidar (CDWL) is proposed. The method employs wavelet covariance transform to locate the SLH and MLH using the local maximum positions and an automatic algorithm of dilation operation. AEC and AOD are determined by the fitting curve using the SNR equation. Furthermore, the method demonstrates the influential mechanism of optical properties on the SLH and MLH. MLH is linearly correlated with AEC and AOD because of solar heating increasing. The results were verified by the data of an ocean island site in China.

## 1. Introduction

The lowest atmospheric layer of the earth is marked by a planetary boundary layer (PBL). There is a variable daily convolution in the structure and composition of the PBL [1]. During the daytime, the PBL is mainly composed of the surface layer (SL), mixed layer (ML), and entrainment zone. During the nighttime, the ML collapses into the nocturnal boundary layer (NBL) and residual layer (RL) [2]. Furthermore, the mixing and residual layers coexist during the sunrise and sunset [1,3]. Atmospheric variables such as potential temperature, aerosol concentration, and specific humidity usually experience sharp gradients at the top of the PBL. Thus, some measurements of PBL height (PBLH) were proposed based on the characteristics of these variables [4]. Additionally, the optical properties including extinction coefficient and optical depth were employed to represent aerosols, including the total amount of pollutants [5,6], which were determined by the size distribution [7,8] of aerosol formation, which was affected by relative humidity (RH) and temperature (T) [9]. There is a complex interaction between PBL height and aerosols and statistical associations between PBL height and levels of pollutants [5,10,11,12,13]. Thus, the PBLH is an important physical quantity for atmosphere environment assessment [14,15,16,17].

These measurement techniques were mainly implemented by microwave radiometer [14], ceilometers [18,19,20], and lidar, including Mie-scattering lidar [21,22] and coherent Doppler wind lidar (CDWL) [23]. The microwave radiometer is based on the thermodynamic properties of the atmosphere for potential temperature and specific humidity [24]. Ceilometers are single-wavelength micro-lidars intended for cloud-base height detection and are ubiquitous in airports and meteorological service centers worldwide [20]. The Mie-scattering lidar employs back-scattering signals to monitor aerosol concentrations [25]. CDWL data are related to the average wind speed [26]. These techniques have been proposed to combine with several algorithms to accurately detect the PBLH based on the sharp gradient. Some algorithms [4] include visual inspection, the threshold method, the gradient method, ideal profile fitting (FIT) [25], wavelet covariance transform (WCT), and variance (or standard deviation) analysis. Many studies have shown that the retrieved PBLH of lidar instruments is in good consistency with the radiometer [27]. However, the accuracy of the PBLH was influenced by multiple-layer aerosol layers and cloud layers [28]. Some methods were proposed to combine some different algorithms, such as combining WCT with the ideal curve-fitting (ICF) algorithm [25], combining WCT with the threshold for a range-corrected signal, and combining WCT with depolarization [3].

The PBL contains aerosols of the low troposphere. The optical properties of aerosols mainly include the aerosol extinction coefficient (AEC) and aerosol optical depth (AOD). It was pointed out that PBLH decreased sharply with the increase of aerosol load [29]. A two-component fitting method is employed to find an accurate AEC as the boundary value in Mie-scattering lidar [30]. However, the boundary value is determined by the empirical back-scattering ratio, which is measured by combining auxiliary sensors, such as a sun photometer [30]. Furthermore, the hundreds of meters of the blind zone and the transition zone in traditional Mie-scattering lidars [31,32] always lead to a difficulty in probing aerosols in the lower troposphere [33], since the biaxial lidars are in parallel to the laser and telescope axes. In addition, CDWL can also be used to estimate the MLH based on the signal-to-noise ratio (SNR) by combining with WCT [23]. However, there are multiple local maximum positions that are manually chosen to determine the PBL. Thus, an automatic PBL extracting algorithm is needed to speed up the determination process.

The existing studies on the interaction between aerosols and the PBL are mainly based on short-term numerical simulations [34] and long-term comprehensive observations [35]. The main influence of aerosols on the PBL is the cooling effect on the surface and the heating effect on the atmosphere. The aerosol extinction in the atmosphere (including the scattering and absorption of sunlight) will reduce the short-wave radiation of the sunlight reaching the surface, so the surface heat flux drives the development of the PBL. In these methods, a lidar and a sunphotometer were synthetically applied to monitor the PBLH, and AOD or AEC, respectively. The AEC and AOD depend on the wavelength of light, and the wavelengths of the sunphotometer and lidar are different. However, no attempt has been made to simultaneously determine the PBLH, AOD, and AEC based on the same lidar.

In this study, the atmospheric boundary layer and the optical properties of aerosols are implemented by employing CDWL and WCT based on two local maximum positions with an automatic algorithm. In this work, the surface-layer height (SLH) and mixed-layer height (MLH) were simultaneously extracted based on wavelet covariance transform with an automatic algorithm, due to the sharp gradient on the boundaries of SL and ML. Meanwhile, the optical properties including AEC were estimated by linear fitting in the range from SLH to MLH, and the AOD was calculated by AEC-times depth. Then, the relationship between optical properties, the SLH, and MLH were quantitatively characterized for an ocean island site in China.

## 2. Materials and Methods

### 2.1. Study Area

The measurements were carried out at the observation site in the ocean island site, which is located in the south of China with a tropical maritime monsoon climate. The weather around the site is summer-like the whole year, the highest temperature is 32 ${}^{{}^{\circ}}$C, and the lowest temperature is 20 ${}^{{}^{\circ}}$C due to the effect of the ocean. The prevailing period of the northeast monsoon is from October to March of the next year, and the prevailing period of the southwest monsoon is from May to September. The rainy period is from June to November and the dry period is from December to May of the next year. Rainless and sunny weather in December was selected as the observation object, and the observation site is far from the city and less affected by emissions from industries, vehicles, and other anthropogenic activities. The aerosols in ocean islands are mainly composed of sea salt aerosols.

### 2.2. Experimental Instruments

The measurements were performed by a CDWL (Windprint S4000, Qingdao Aerospace Seaglet Environmental Technology Ltd., Qingdao, Shandong, China), whose technical specification is shown in Table 1. The vertical resolution and temporal resolution of this CDWL are 30 m and 1 s, respectively. The telescope was designed with a diameter of 40 m and a focal length of 1000 m. The blind zone of CDWL is 60 m. The typical SNR image, which includes successive 180 measurements, is shown in Figure 1a. The SNR of one measurement and the average SNR of the successive 180 measurements are demonstrated in Figure 1b. The PBL is in the range of red rectangular area and the AEC of PBL is homogeneous. This work presents an automated algorithm to simultaneously extract the PBLH and AEC. The weather in December was chosen for typical case to verify the feasibility of the proposed method. The continuous sample data of 24 h by the CDWL was used to study the daily evolution of PBLH and the optical properties of the aerosol.

### 2.3. SNR of Coherent Doppler Wind Lidar (CDWL)

The SNR of the CDWL mainly depends on four factors: the average direct detection power, the heterodyne efficiency, the wavelength λ, and the receiver bandwidth *B* [36]. Under the conditions of negligible refractive-turbulence effects, the matched filter $B=\frac{1}{\tau }$, where τ is the pulse duration and far-field operation, the peak of SNR depends on the altitude *z*, and can be expressed as [37]:(1)$$SNR\left(z\right)=\frac{\pi \eta_{Q}U_{T}\lambda \beta D^{2}{\left[T\left(z_{m}\right)\right]}^{2}}{8hBz^{2}}\propto \frac{{\left[T\left(z_{m}\right)\right]}^{2}}{z^{2}}$$
where $\eta_{Q}$ is the quantum efficiency of the detector, *h* is the Planck constant, $U_{T}$ is the transmitting pulse energy, β is the back-scattering coefficient, and *D* is the diameter of laser beam. $T\left(z_{m}\right)=exp\left[-\int_{0}^{z_{m}}\alpha \left(r\right)dr\right]$ is the dimensionless one-way irradiance extinction at wavelength λ, and $\alpha (m^{-1})$ is the linear AEC along the propagation path. Figure 1b shows the typical SNR in terms of altitude *z*.

### 2.4. WCT

The Haar wavelet is discontinuous and usually applied to the location of the PBL due to its superior spatial location and computational efficiency. The Haar wavelet function can be expressed as:(2)$$h\left(\frac{z-b}{a}\right)=\left\{\begin{array}{cc} -1 & b-\frac{a}{2}\leq z\leq b\\ +1 & b\leq z\leq b+\frac{a}{2}\\ 0 & otherwise \end{array}\right.$$
where *z* is the altitude, *a* is the dilation of the function, and *b* is the center of the Haar function. The Haar wavelet function is shown in Figure 2a and the WCT of the Haar function is defined using Equation [38]:(3)$$W_{f}\left(a,b\right)=a^{-1}\int_{z_{b}}^{z_{t}}f\left(z\right)h\left(\frac{z-b}{a}\right)dz$$
where $z_{t}$ and $z_{b}$ are the spatial ranges in the profile, f(z) is the profile as a function of altitude and the normalization factor, and $a^{-1}$, is the inverse of the dilation. The first step in the algorithm to determine the PBLH is to define the dilation of the Haar function values. Figure 2b indicates the WCT of SNR with different dilation. The minimum of WCT was chosen as an objective parameter to find its optimal value of dilation. Figure 2c shows the minimum of WCT dependent on the dilation, and the position of the minimum value was chosen as the appropriate dilation for Haar function. The corresponding dilation is 60 m.

The WCT was applied to the profile with the dilation of 60 m for the Haar function. Figure 3 demonstrates that the position $P_{a}$ is identified by the local minimum value in the resulting wavelet covariance profile and indicates the height of the strongest decrease of SNR. The altitude of 180 m can be considered as the SLH that is larger than the blind zone of 60 m. $P_{b}$ is determined by the local minimum value in absolute value of $W_{f}\left(a,b\right)$, which means the local minimum value of SNR, and the height could be seen as the MLH with an altitude of 840 m. The SLH and MLH are consistent with the results in reference [2]. The local maximum positions of absolute WCT can be automatically determined by dilation operation, which is defined as $I⊕E=\;\max_{b\in E}\left[I\left(x+b\right)-E\left(b\right)\right]$, where *I* represents the signal and *E* denotes the structuring element [39]. The dilation operation has a filtering effect that suppresses dark regions smaller than structuring elements and results in the enlargement of bright ones. The dilation operation can be recast into maximum operation on structuring elements.

### 2.5. AEC and AOD

The aerosols in the atmosphere in the range from $P_{a}$ to $P_{b}$ can be seen as roughly randomly distributed particles in PBL, and the corresponding linear extinction coefficients can be regarded as homogeneous [40]. Thus, the irradiance extinction *T* can be given by $T=exp\left[-\alpha \left(P_{b}-P_{a}\right)\right]$ at the PBL. The linear extinction coefficient α can be obtained by the fitting curve of Equation (Equation 1) when the boundaries of layers are obtained by local minimum values in the resulting wavelet covariance profile. Equation (Equation 1) made the logarithmic transform and can be expressed as:(4)$$log\left[SNR\left(z\right)\right]=-2\alpha \left(r\right)z-2logz$$

Furthermore, the corresponding AOD at the wavelength of 1550 nm is defined by [35]:(5)$$AOD=\alpha \left(r\right)\left(P_{b}-P_{a}\right)$$

To sum up, Figure 4 demonstrated the flowchart to determine the four parameters including SLH, MLH, AEC and AOD.

## 3. Results

Figure 5 demonstrates that the typical SLH and MLH, which are extracted from the mean of 180 measurements of SNR, depend on the local time during the whole day. During the daytime, the PBLH is identical to the MLH. During the nighttime, the MLH collapses into the nocturnal boundary layer (NBL) and residual layer (RL). The MLH is identical to the height of the NBL. In addition, Figure 5a,b indicate that the MLH is negatively correlated with AEC and positively correlated with AOD in terms of the local time. Figure 5c,d demonstrate that the linear fitting curves of the MLH depending on AEC and AOD can be expressed as: $MLH=K_{1}\times AEC+C_{1}$, and $MLH=K_{2}\times AOD+C_{2}$, where $K_{1}$ and $K_{2}$ are constants, and $C_{1}$ and $C_{2}$ denote constants which do not affect the result. Their correlation coefficients *R* are 0.67 and 0.65, respectively. The linear functions of the SLH dependent of AEC and AOD are given by: $SLH=K_{3}\times AEC+C_{3}$, and $SLH=K_{4}\times AOD+C_{4}$, where $K_{3}$ and $K_{4}$ are positive constants, and $C_{3}$ and $C_{4}$ are constants. However, their correlation coefficients *R* are relatively small, and the values of *R* are 0.51 and 0.16, respectively.

MLH is linearly correlated with AEC and AOD, and Figure 6a demonstrates that the slopes $K_{1}$ of the MLH dependent on AEC are negative, and the slopes of $K_{2}$ of the MLH linearly dependent on AOD are positive, which means that the MLH decreases while the AEC is increasing, and the MLH increases while the AOD is increasing. The reason is that solar heating increases in the ML while the strength of capping inversion decreases, leading to a rise in the MLH and decrements in AEC. There is a positive relationship between the MLH and AOD and a negative between MLH and AEC. The difference is that the effect of increment of MLH on AOD is greater than that of the decrement of AEC. Thus, the effect that solar heating increases in the MLH is greater than the effect of MLH on AEC.

SLH is linearly correlated with AEC and AOD, and Figure 6b shows the distribution of the slopes $K_{3}$ and $K_{4}$ in eight successive days. The values are sometimes positive and sometimes negative, which means that the linear fitting curves of SLH dependent on AEC and AOD are complex. The reasons are the multiple factors such as the cooling effect of the surface enhanced with the increase of AOD and aerosols with human activity.

In order to study the factor of aerosols with different sizes on AEC, the data of $PM_{2.5}$ and $PM_{10}$ are obtained from the National Urban Air Quality data of the Ministry of Ecology and Environment, PRC [41]. Figure 7a indicates the positive correlation between AEC and aerosols ($PM_{2.5}$ and $PM_{10}$) during the local time. The Pearson correlation coefficient provides a measure of the strength of the linear association between two variables [42], and it is found that the correlation coefficient between the derived AEC and aerosols ($PM_{2.5}$ and $PM_{10}$) are 0.1026 and 0.5890, which suggested that aerosol of $PM_{2.5}$ plays an important role in the determination of AEC. Additionally, Figure 7b demonstrates that there are positive statistical associations between AEC and the mean of wind speed, which is estimated by the same CDWL. However, AOD is not positively related to the mean wind speed. Thus, the factors considered for AEC are much simpler than AOD.

The comparison of the AEC with the optical absorption coefficient (OAC) is based on photoacoustic spectroscopy at the wavelength of 1064 nm [43], and it is found that the trend of the AEC is highly correlated with the OAC, shown in Figure 8a. Furthermore, the reference data of AOD and MLH were obtained from EAC4 (ECMWF Atmospheric Composition Reanalysis 4) [44], which is the fourth generation ECMWF global reanalysis of atmospheric composition, and reanalysis combines model data with observations from across the world into a globally complete and consistent dataset using a model of the atmosphere based on the laws of physics and chemistry. Figure 8b,c demonstrate that the trends of AOD and MLH are related to that in EAC4. Thus, it is feasible that the simultaneous extraction method of the planetary boundary-layer height and aerosol optical properties can be obtained from coherent Doppler wind lidar.

## 4. Discussion

AEC is the result of both absorption and scattering [40]: $a_{ext}=n\left(C_{abs}+C_{sca}\right)$, where *n* is the number of particles per unit volume, and $C_{abs}$ and $C_{sca}$ are the absorption and scattering cross-sections, respectively. The light with a wavelength of 1550 nm passing through aerosols is attenuated almost entirely by scattering. The scattering cross-section depends on the size of the aerosols. It was found that the vertical meteorological parameters, such as relative humidity and temperature, and the aqueous and heterogeneous atmospheric chemical reactions altogether led to the aerosol formation [9] and resulted in different size distributions. Additionally, other parameters such as wind, rainfall, and even the emission rates will change the number of particles in unit volume *n* in the physical view. Thus, the AEC is affected by many parameters.

In this study, the signal-to-noise ratio of CDWL had been used to simultaneously extract four parameters, including SLH, MLH, AEC, and AOD, which simultaneously monitor the daily evolution of both the PBL height and the optical properties of aerosols and their relationships. Although the interaction between the aerosols and the PBL height is highly complicated, there is a positive relationship between MLH and AOD, and negative with AEC, which suggests that the effect of the increment of MLH on AOD is greater than that of the decrement of AEC. Thus, the effect that solar heating increases in the MLH is greater than the effect of MLH on AEC.

In this work, CDWL was used for measuring both PBLH and optical properties, since the system has a smaller blind zone than traditional Mie-scattering lidar due to the coaxial design of CDWL with the telescope axis. In addition, SLH can be extracted by the CDWL, which is difficult to estimate in Mie-scattering lidar.

Ruijun Dang, et al. [4] had made a review of techniques for measuring the atmospheric boundary-layer height (ABLH) or the MLH using aerosol lidar. In their review, many studies on measurements of ABLH were based on range-corrected SNR (RCSNR). The RCSNR can be obtained by Equation (Equation 1) multiplying $z^{2}$, which can be expressed as [4]:(6)$$RCSNR\left(z\right)\propto T\left(z_{m}\right)=exp\left[-\int_{0}^{z_{m}}\alpha \left(r\right)dr\right]$$

Classical WCT methods were also applied for extracting the ABLH or MLH. When the Haar wavelet function *h* encounters a sharp drop in RCSNR, a local maximum in $W_{f}\left(a,b\right)$ occurs, indicating a step change in the RCSNR located at *b* with a coherent scale of *a*. Therefore, the ABLH is defined as the location of *b*, where the $W_{f}\left(a,b\right)$ reaches its maximum.

Figure 9 shows the RCSNR and the corresponding WCT and demonstrates that the local maximum of WCT of RCSNR is at the altitude of 120 m. It is lower than 180 m, as shown in Figure 3. Thus, the local maximum of WCT of RCSNR is the location of SLH, which is consistent with the classical WCT method. However, the classical method cannot obtain the MLH, and the AEC between the SL and ML cannot be estimated. In order to overcome it, the local minimum values of WCT based on RCSNR are employed at the location of altitude of 900 m, which is smaller than the 840 m extracted by our algorithm. Therefore, the MLH can be defined as the local minimum values of the WCT of RCSNR.

In addition, the cloud has a strong effect on the accurate extraction of PBLH, and the opening filter, which is defined as the two sequential compositions of erosion and dilation, can be first applied to the SNR image to reduce the cloud before the mean of 180 measurements of SNR. Figure 10a shows that the bright spots are the clouds due to the strong scattering, and the clouds can be filtered with an opening operation, as shown in Figure 10b.

## 5. Conclusions

In this study, a method of simultaneously extracting the SLH, MLH, and optical properties based on the SNR of the same CDWL was presented. The method employed WCT to locate the SLH and MLH, and optical properties including AEC and AOD were determined by the fitting curve using the SNR equation. In addition, the effects of optical properties on the SLH and MLH were qualitatively studied for an ocean island site in China. The results preliminarily demonstrated that MLH is linearly correlated with AEC or AOD because of increasing solar heating. Furthermore, there is a positive relationship between MLH with AOD and negative with AEC, which suggests the effect that solar heating increases in the MLH are greater than the effect of MLH on AEC. However, the effect of optical properties on SLH is complex. Thus, this work provides an effective method for understanding the aerosol effect on PBL in the same location.

## Figures and Tables

**Figure 1 sensors-22-03412-f001:**
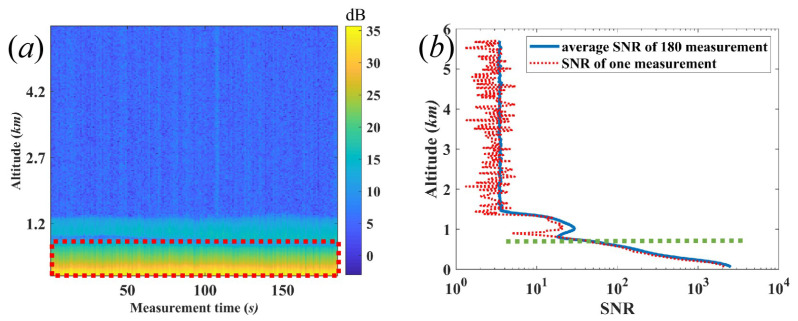
(**a**) SNR image of successive 180 measurements, (**b**) SNR of one measurement, and the average SNR of 180 successive measurements. The PBL is in the red rectangular area. The green dashed line denotes the top of PBL, which can be considered as the MLH.

**Figure 2 sensors-22-03412-f002:**
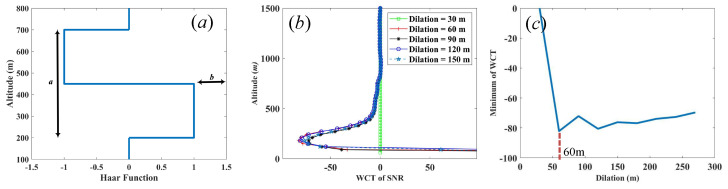
(**a**) Plot of the Haar wavelet function, (**b**) WCT of SNR at the different dilation, and (**c**) the minimum of WCT depending on dilation.

**Figure 3 sensors-22-03412-f003:**
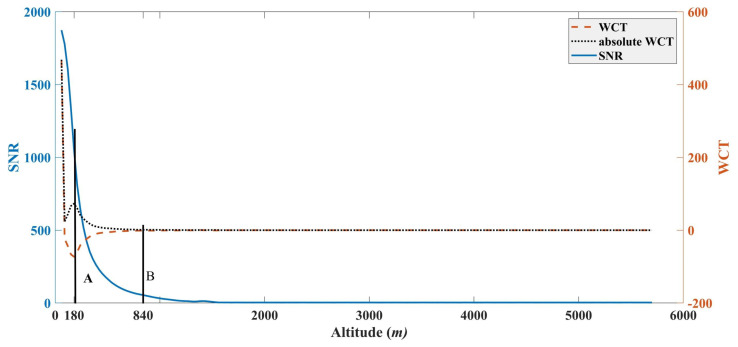
SNR and WCT of SNR in terms of altitude. A and B denote the local minimum values of WCT of SNR and the absolute value of WCT of SNR, and the corresponding altitudes are labeled with $P_{a}$ and $P_{b}$, respectively.

**Figure 4 sensors-22-03412-f004:**
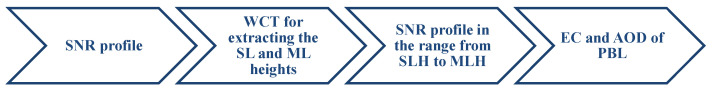
Flowchart for determination of SLH/MLH and its AEC/AOD.

**Figure 5 sensors-22-03412-f005:**
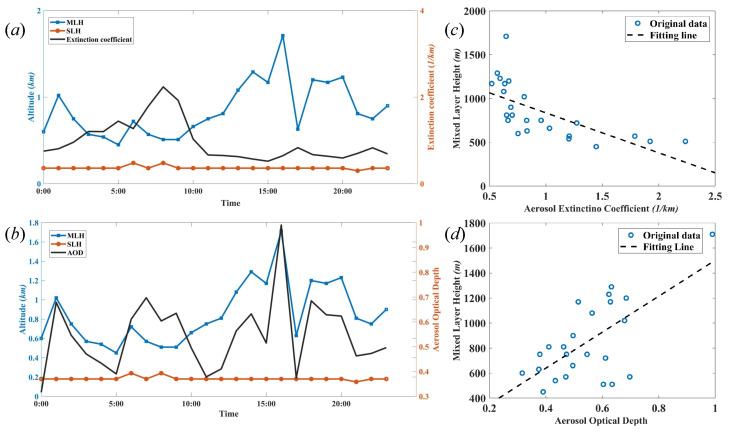
(**a**) SLH, MLH, and AEC during local time, and (**b**) SLH, MLH, and AOD during local time. MLH is a linear relationship with (**c**) AEC and (**d**) AOD.

**Figure 6 sensors-22-03412-f006:**
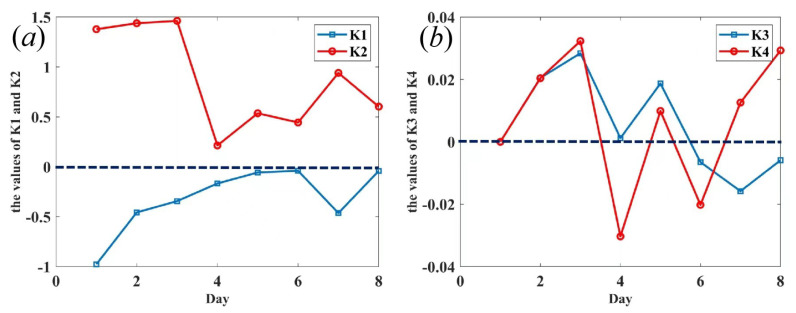
(**a**) Slope of MLH depending on AEC and AOD, (**b**) slope of SLH depending on AEC and AOD for successive 8 days.

**Figure 7 sensors-22-03412-f007:**
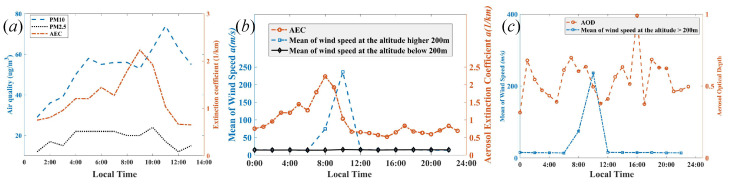
(**a**) The correlation between AEC and aerosols with different sizes including $PM_{2.5}$ and $PM_{10}$ during the local time, (**b**) statistical associations between AEC and mean of wind speed, and (**c**) AOD and mean of wind speed during local time.

**Figure 8 sensors-22-03412-f008:**
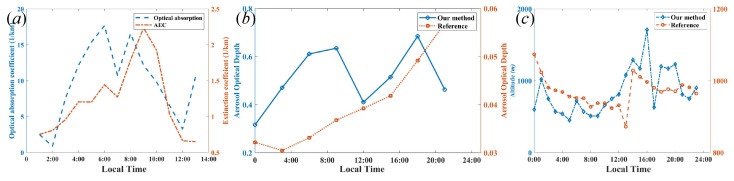
(**a**) The trend of the AEC and optical absorption coefficient, (**b**) the correlation between AEC and air quality, including $PM_{2.5}$ and $PM_{10}$ during the local time, and (**c**) the MLH correlation during the local time.

**Figure 9 sensors-22-03412-f009:**
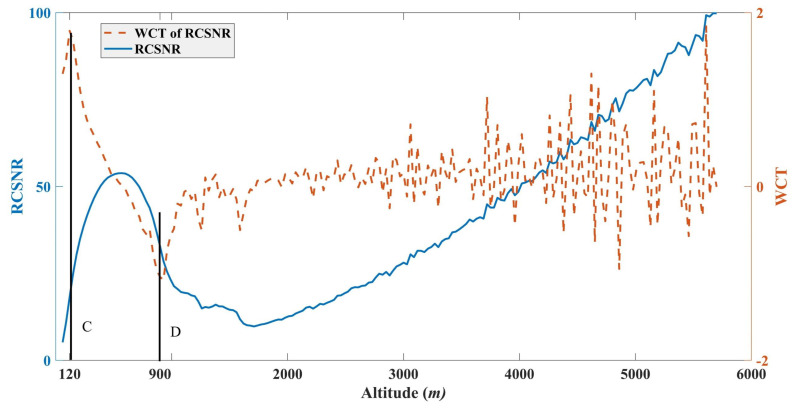
RCSNR and WCT of RCSNR in terms of altitude. C and D denote the local maximum and minimum values of WCT of RCSNR.

**Figure 10 sensors-22-03412-f010:**
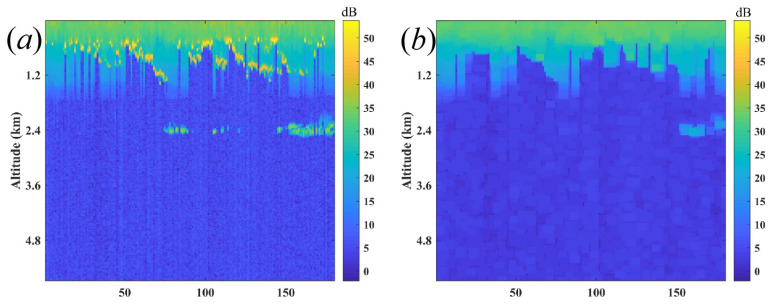
(**a**) Lidar image destroyed by the clouds, (**b**) filtered lidar image with the morphological opening operation.

**Table 1 sensors-22-03412-t001:** Technical specifications of Windprint S4000.

Parameter/Unit	Value
Wavelength/nm	1550
Pulse repetition rate/kHz	10
Pulse energy/uJ	≥150
Pulse width/ns	100
Power consumption/W	<300

## Data Availability

The data presented in this study are available on request from the corresponding author.

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
