# Peer review of "Simultaneous Extraction of Planetary Boundary-Layer Height and Aerosol Optical Properties from Coherent Doppler Wind Lidar"

_sensors, 2022, doi:10.3390/s22093412_

Round 1

Reviewer 2 Report

    This study reported a method of simultaneously extracting the surface layer height (SLH), mixed layer height (MLH), and optical properties based on the signal-to-noise ratio (SNR) of the same coherent Doppler wind lidar (CDWL). The SLH and MLH are found using the wavelet covariance transform method. The aerosol extinction coefficient (AEC) and aerosol optical depth (AOD) are determined by the fitting curve using the SNR equation. The research method has been described clearly. Some points still need to be further addressed. I would suggest this paper be major revision.  

Major comments

  1. The data (chapter 2) need to be further elaborated. Some detail of data properties should be addressed, such as the observation period or data length, observation frequency or observation time, the weather conditions (sunny, cloudy, or rainy), data sample size, etc... The PBL, MLH, or aerosol properties are highly influenced by weather conditions. The observation frequency and observation time are important since chapter 3 is discussing the diurnal evolution of PBLH. The readers now could not know the applicability of this method or data.
  2. I also suggest the authors explain why and how the analysis is made in the results section, Figure 5–8. Are the results based on some specific cases or a long-time average? Does the 8-days analysis in figure6 represent classical cases?
  3. This method is studied for a mid-latitude coastal area in China. I would suggest the authors add some discussion for the limitation of this method in other areas or locations. What geographical factors might affect this method?            

Minor comments

Figure1a: How do you determine the range of red rectangular (the PBL)? Figure1b: Add the description for green dashed line in caption.

Reviewer 3 Report

In lines 133 to 136 refering to figure 6, it is said "that the slopes K1 of MLH dependent on AEC are negative, and the slopes of K2 of MLH linearly dependent on AOD are positive".
It is very difficult to judge at glance which slopes are positive or negative.

Reviewer 4 Report

This study presents the extraction of MLH and aerosol optical properties (extinction coefficient and AOD) in a site in east China by means of a Doppler wind lidar. The paper is really short for the issue of boundary-layer dynamics and associated air pollution, the analysis is really poor with few figures indocating a diurnal pattern, which is not clear if it refers to a single day or it's averaged over a period of measurements. The literature overview is really poor also. The discussion section also contains results and figures, it's not considered as a real discussion. Overall, the manuscript gives the sense of a quick-written work with minimal analysis, which cannot be published in the journal. Analytic comments, edits and suggestions for a possible revision or new study are included in the attached pdf file. 

Round 2

Reviewer 2 Report

The authors have fully considered the comments from the first review. The modified manuscript is improved and elaborated according to the comments. The descriptions are more clear compared with the previous version. I suggest accepting it in its present form.

Author Response

Thank you very much for providing such positive and encouraging comments and valuable suggestions.

Reviewer 4 Report

Despite the fact that I was negative in my original review of this paper, authors improved in substantially, and indeed, they have done a great work on it, by enriching the literature, providing physical explanations of the main findings, clarifying some parts of the methodology and in general, improve the revised manuscript. I think that the current one can stand as a research paper, mostly technical one, in the journal and I recommend its publication.

Author Response

Thank you very much for reading the paper carefully and providing valuable comments and suggestions. We have sent our paper to a native English speaker whose research is specialized in this field to check the English. We have made a careful check and a substantial revision of the paper, and have simplified the sentence structures and corrected grammatical errors.